

# The roles of trait and process resilience in relation of BIS/BAS and depressive symptoms among adolescents

Akihiro Masuyama[1], Takahiro Kubo[2], Hiroki Shinkawa[3] and Daichi Sugawara[4]

[1] Psychology, Iryo Sosei University, Iwaki, Fukushima, Japan
[2] Human Sciences, University of Tsukuba, Bunkyo-ku, Tokyo, Japan
[3] Education, Hirosaki University, Hirosaki, Aomori, Japan
[4] Human Sciences, University of Tsukuba, Tsukuba, Ibaraki, Japan

## ABSTRACT

**Background:** Extensive literature revealed the relations of depression with behavioral inhibition system (BIS) and behavioral activation system (BAS) as vulnerability and with resilience separately. Besides, the concept of resilience is still broad and ambiguous. Thus, this study aimed to reveal the mediation of two aspects of resilience: trait and process, in the relations of BIS and BAS to depression among adolescents.

**Methods:** The data set used in this study was a cross-sectional survey among 965 adolescents. The obtained data from the self-reported questionnaires used in this study were as below: Depression Serf-Rating Scale for Children (DSRS-C), Behavioral Inhibition and Behavioral Activation System Scale (BIS/BASS), and Bidimensional Resilience Scale (BRS). Structural equation modeling (SEM) was conducted to verify the hypothesized relations among BIS/BAS, trait and behavior resilience, and depressive symptoms.

**Results:** The obtained indices of fit from SEM were good or sufficient ($\chi^2$ = 562.911, $df$ = 96, $p$ < 0.001; CFI = 0.925; NFI = 0.913; RMSEA = 0.073, 90% CI [0.067, 0.079]; SRMR = 0.066). And the modeling showed that both BIS/BAS directly and indirectly influenced to depression. The indirect effects of BIS/BAS were mediated only trait resilience except the indirect effect of BIS *via* behavioral resilience.

**Discussion:** Our results suggested that trait resilience played a significant mediation role in the relationships between BIS/BAS and depression. Trait but not process resilience could be considered suitable as an intervention target in line with decreasing depression.

# INTRODUCTION

Depression is a global mental health concern. According to the World Health Organization (*World Health Organization, 2021*), more than 260 million people worldwide are affected by depression. For depression among the young population, a meta-analysis has estimated a 2.6% prevalence rate of any depressive disorder (*Polanczyk et al., 2015*). Depressive symptoms during adolescence cause negative outcomes and serve

Corresponding author
Akihiro Masuyama,
ak.masuyama@gmail.com

as a risk factor for the onset of depressive disorder in adulthood (*Fogel, Eaton & Ford, 2006*; *Johnson et al., 2018*).

Extensive literature has revealed the significance of psychological resilience as a protective factor against depression in adolescents (*Anyan & Hjemdal, 2016*; *Edward, 2005*; *Hjemdal et al., 2011*; *Ng, Ang & Ho, 2012*). Psychological resilience refers to "the process of effectively negotiating, adapting to, or managing significant sources of stress or trauma" (*Windle, 2011*), and includes various psychological and social factors. In line with the relationship between depression and psychological resilience, *Ding et al. (2017)* found that resilience related to stress coping has a buffering effect on depressive symptoms with childhood trauma. *Skrove, Romundstad & Indredavik (2013)* also showed an association between depression and psychological resilience by suggesting a pathway in which higher psychological resilience leads to good interpersonal relationships that follow lower depression.

In psychology, studies on resilience have noted the tendency for the concept to fall into ambiguity. *Southwick et al. (2014)* called for the need to build an integrative view and definition. The concept of resilience is complexly constructed by biological, psychological, social, and cultural factors and their interactions. In psychology, the construct of resilience has been accounted for in various ways: trait, process, and outcome (*Fletcher & Sarkar, 2013*). *Ahern (2006)* pointed out the development of psychological scales for measuring resilience has resulted in a broad and ambiguous definition. The roles of resilience suggested by previous studies can be well explained by organizing the concept of resilience based on the distinction between trait, process, and outcome (*Fletcher & Sarkar, 2013*). Accordingly, psychological resilience can be divided into either trait or process aspects. However, research has not provided empirical findings on this method of organizing itself. Investigations have yielded results on the effect of resilience from the single view of either trait or process. Still, no empirical studies have demonstrated the comparing among trait aspect and the process aspect of resilience. Trait resilience could be assumed to affect mental health problems, including depressive symptoms, *via* process resilience. In addition to the trait–process resilience pathway, the model remains unclear. Thus, the current study used the two aspects of trait and process (behavioral) to define the concept of resilience and then investigated their relation to depression.

Contrary to resilience as a concept of protective factor, a large body of evidence shows the behavioral inhibition system (BIS) and behavioral activation system (BAS; *Carver & White, 1994*) as vulnerabilities for depression. The BIS/BAS theory based on biology has been used to explain personality construction. According to the reinforcement sensitivity theory (*Gray, 1987*), personality is primarily composed of two motivational systems (BIS and BAS). The BIS is hypothesized to be sensitive to threats and punishment, thus involving avoidance and negative affect. The BAS is hypothesized to be sensitive to rewards and drive, thus involving approach and positive affect (*Zinbarg & Lira Yoon, 2008*; *Erdle & Rushton, 2010*). Both BIS and BAS are associated with depression (*e.g.*, *Kasch et al., 2002*; *Pinto-Meza et al., 2006*; *Mellick, Sharp & Alfano, 2014*): a higher BIS and a lower BAS are related to depressive symptoms. That is, depression is characterized by high sensitivity toward threats, increased avoidant behavior and negative affect, slight sensitivity

toward rewards, and decreased approach behavior and positive affect. As the BIS/BAS was thought to be a temperature based on the primary construction of personality, the BIS/BAS has been studied in terms of its relationship with other aspects of trait. *Slobodskaya (2007)* found correlations between the BIS/BAS and Big Five personality traits; BIS is correlated with neuroticism and extraversion, whereas BAS is correlated with agreeableness and conscientiousness. *Włodarska et al. (2021)* also showed the correlations between BIS/BAS and dark triad personality traits through meta-analysis, including positive correlations between Narcissism, Machiavellianism, and BAS. Although BIS/BAS and psychopathology and personality have been broadly investigated, little is known about their relation to resilience.

Several studies revealed the relationships between BIS/BAS and some positive psychological aspects in relation to resilience. Regarding optimism, which has been associated with resilience (*Souri & Hasanirad, 2011*), *De Pascalis et al. (2013)* revealed that BIS and optimism had a commonality in brain activity through electroencephalography and a negative correlation between self-measured BIS and optimism. *Taubitz, Pedersen & Larson (2015)* investigated the relationships between BIS/BAS and adaptive psychological traits, such as cognitive reappraisal. The results showed that reward responsiveness, a subfactor of BAS, could predict cognitive reappraisal and well-being (*Taubitz, Pedersen & Larson, 2015*). Based on the results that BIS/BAS could influence traits and that BIS/BAS was an entire brain system that affects various psychological aspects, it was assumed that BIS/BAS also affects trait resilience. In addition to these pathways, it would also be considered that trait resilience leads to behavioral aspects of resilience and then protect against depression. However, no empirical study has investigated the relation between BIS/BAS, resilience, and depression.

Thus, we aimed to investigate the relations among BIS/BAS, depression, trait and process aspects of resilience. For distinction of trait and process resilience, we considered process resilience as behavioral resilience according to the characteristics of the scale used in this study (detail was described in Measure section). Based on Gray's theory and previous related studies, we assumed that BIS/BAS would primarily drive trait and behavioral resilience. Therefore, we hypothesized the pathways by which BIS/BAS would affect traits and resilience and then influence depression, as depicted in Fig. 1. We tested these pathways to examine the mediation effects of trait and behavior resilience and their associations.

## MATERIALS AND METHODS

### Data set

We used the 2021 dataset of the I'M HAPPY project in Japan, launched in 2019. This project aimed to investigate psychological factors and mental health among adolescents aged 12 to 15 through longitudinal surveys in two junior high schools in Japan (*Masuyama et al., 2021a*; osf.io/4dfb8).

The survey for the dataset of 2021 was carried out during the COVID-19 pandemic; in July 2021, the number of confirmed cases of infection per day was about 30 in the area where the survey was conducted: Fukushima Prefecture, Japan. The dataset of 2021
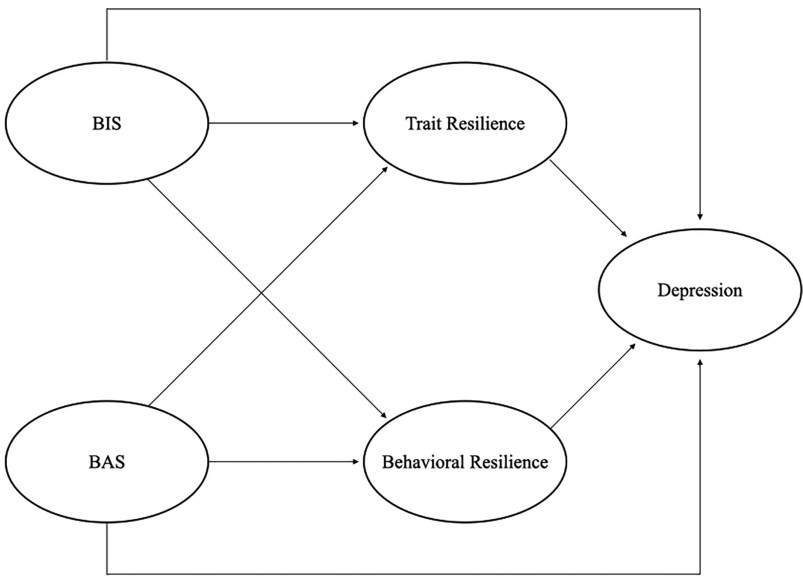

**Figure 1 Hypothesized model.** BIS, Behavioral Inhibition System; BAS, Behavioral Activation System.

contained responses on depressive symptoms, trait anxiety, sleep disturbance, fear of COVID-19, internet addiction symptoms, and psychological resilience from 965 adolescents (mean age = 14.03 years, *SD* = 0.67). The current study used the measures of depressive symptoms, BIS/BAS, and resilience. Informed consent was obtained from the board of education, school, and teachers. Informed consent from participants was obtained by providing their responses to the questionnaire. The overall procedure for this longitudinal survey was approved by the ethics committee of Iryo Sosei University (Receipt number: 21-09).

## Measure

### Depression serf-rating scale for children (DSRS-C)

To assess depressive symptoms, the Japanese version of the DSRS-C (*Birleson, 1981*) was used, an 18-item scale that includes five reverse items. Participants responded on a three-point Likert ranging from "0" (no/never) to "2" (most of the time). The Japanese DSRS-C has good validation and reliability (*Murata et al., 1996*). The total score ranges from 0 to 36, with a higher score indicating more significant depressive symptoms. The scale's internal consistency in this study was $\alpha = 0.865$.

### Behavioral inhibition and behavioral activation system scale

To assess participants' BIS and BAS, the Japanese version of the BIS/BAS Scale (*Carver & White, 1994*), which has 20 items to which participants responded using a four-point Likert scale ranging from "1" (totally disagree) to "4" (totally agree), was used.
The BIS/BAS scale has a four-factor construction: (1) BIS, (2) BAS Reward Responsiveness, (3) BAS Drive, and (4) BAS Fun Seeking. The total scores of each factor are calculated and then used in statistical analyses. The Japanese BIS/BAS scale has been confirmed

reliable and valid (*Takahashi et al., 2007*). The internal consistency in this study was as follows: (1) BIS: $\alpha = 0.706$, (2) BAS Reward Responsiveness: $\alpha = 0.831$, (3) BAS Drive: $\alpha = 0.846$, and (4) BAS Fun Seeking: $\alpha = 0.758$.

### Bidimensional resilience scale (BRS)

Psychological resilience was measured using the BRS (*Hirano, 2010*), a 21-item scale with two dimensions: innate and acquired resilience. The innate resilience dimension includes four factors (optimism, control, sociability, and vitality). Example items are "I can overcome difficult events" (optimism factor) and "Putting in effort is important" (vitality factor). The acquired resilience dimension includes three factors (solving a problem, self-understanding, and understanding others). Example items are "When a problem occurs, I collect information to solve that problem" (problem-solving factor) and "I understand my personality well" (self-understanding factor). Participants responded on a five-point Likert scale ranging from "1" (strongly disagree) to "5" (strongly agree).

The BRS was developed in Japan based on the Temperament/Character model (*Cloninger, Svrakic & Przybeck, 1993*) to classify resilience-related items into the innate and acquired dimensions separately. We regarded the innate and acquired dimensions to indicate trait and process resilience. The reason is that the items in the innate dimension of the scale refer to the participants' ability and trait, and those in the acquired dimension, to participants' actual behavior and response. The BRS has good reliability and validity (*Hirano, 2010*). In this study, the internal consistency values were as follows: trait resilience, $\alpha = 0.896$; process resilience, $\alpha = 0.806$; and total, $\alpha = 0.911$.

## Statistical analyses

The missing values were replaced from the dataset with the mean of the corresponding items. We computed all variables separately for each subscale and factor. Our study used IBM SPSS Statistics for Windows, Version 26.0, to calculate the descriptive statistics and correlations. Relations among variables were evaluated using Pearson's correlation.

Our study conducted structural equation modeling (SEM) to verify the hypothesized relations among BIS/BAS, trait and behavior resilience, and depressive symptoms, using AMOS version 26. The overall hypothesized model is as follows: each factor of BIS/BAS affects the trait and behavioral aspects of resilience, trait resilience also affects behavior resilience, and behavior resilience affects depressive symptoms. We also estimated each factor's direct and indirect effects on depressive symptoms. To evaluate the hypothesized model, we calculated the goodness-of-fit indices of the SEM. Specifically, we used the comparative fit index (CFI), normed fit index (NFI), root mean square error of approximation (RMSEA), and Standard Root Mean Square Residual (SRMR) (*Bentler, 1990*; *Steiger, 1998*). CFI and NFI values above 0.90 suggest a good fit (*Hu & Bentler, 1999*). For RMSEA and SRMR, values below 0.05 indicate good model fit, whereas values above 0.10 indicate unacceptable fit (*Browne & Cudeck, 1993*; *Hu & Bentler, 1999*).

To evaluate the mediation effect of trait and behavioral resilience, in addition to model testing, the mediation analyses were conducted with the model 4 setting of the PROCESS macro developed by *Hayes (2013)*. We applied the 95% bias-corrected

**Table 1 Descriptive statistics (*M* (*SD*)).**

| | Female (*n* = 470) | | | Male (*n* = 493) | | |
|---|---|---|---|---|---|---|
| | Grade 1 | Grade 2 | Grade 3 | Grade 1 | Grade 2 | Grade 3 |
| Depression | 12.57 (6.60) | 12.98 (6.87) | 12.35 (6.24) | 9.58 (5.74) | 10.22 (5.41) | 9.82 (5.80) |
| BIS | 19.06 (3.84) | 19.39 (3.90) | 19.69 (3.17) | 16.44 (4.64) | 17.01 (4.60) | 17.28 (4.09) |
| BAS drive | 10.78 (3.02) | 10.79 (2.95) | 11.19 (2.72) | 10.82 (3.63) | 10.56 (3.07) | 11.33 (2.98) |
| BAS reward responsiveness | 14.89 (3.70) | 15.01 (3.58) | 16.03 (3.08) | 14.36 (4.08) | 14.48 (3.65) | 15.38 (3.47) |
| BAS fun seeking | 10.07 (2.90) | 10.23 (2.66) | 10.86 (2.43) | 10.02 (3.14) | 10.3 (2.81) | 10.81 (2.75) |
| Trait resilience | 37.9 (10.64) | 38.07 (9.45) | 39.15 (8.91) | 40.15 (11.21) | 40.34 (9.36) | 40.43 (9.62) |
| Behavioral resilience | 30.51 (6.29) | 30.52 (5.83) | 32.26 (5.78) | 30.13 (7.27) | 31.06 (6.68) | 31.52 (6.45) |

**Note:**
 "Grade" means the grade in junior high school. BIS, Behavioral Inhibition System; BAS, Behavioral Activation System.

**Table 2 Correlations between each variable.**

| | I | II | III | IV | V | VI | VII |
|---|---|---|---|---|---|---|---|
| I. Depression | – | | | | | | |
| II. BIS | 0.225* | – | | | | | |
| III. BAS drive | −0.247* | 0.184* | – | | | | |
| IV. BAS reward responsiveness | −0.263* | 0.276* | 0.547* | – | | | |
| V. BAS fun seeking | −0.162* | 0.244* | 0.433* | 0.607* | – | | |
| VI. Trait resilience | −0.610* | −0.078 | 0.321* | 0.357* | 0.312* | – | |
| VII. Behavioral resilience | −0.395* | 0.113* | 0.354* | 0.420* | 0.293* | 0.661* | – |

**Notes:**
 * $p < 0.001$.
 BIS, Behavioral Inhibition System; BAS, Behavioral Activation System.

confidence interval obtained from 10,000 bootstrap resampling. The significance of mediation effects was tested by determining whether the confidential interval did not pass by zero or not.

## RESULTS

### Descriptive statistics and correlational analysis

Table 1 shows the descriptive statistics separated by gender and grade. Table 2 also shows the correlations among each variable. The correlation analysis showed that depression was significantly correlated with each factor of BIS/BAS ($r > 0.162$, $p < 0.001$) and trait and behavioral resilience ($r > 0.395$, $p < 0.001$). Significant correlations were also found among BIS/BAS factors and trait and behavioral resilience ($r > 0.113$, $p < 0.001$), but not between BIS and trait resilience ($r = -0.078$, $p = 0.016$).

### Structural equation modeling

To test the hypothesized model that the BIS and BAS would predict depressive symptoms directly and indirectly *via* trait and behavioral resilience, we conducted SEM. The model contained five latent variables: BIS, BAS, trait resilience, behavioral resilience, and depression. For the latent variables of BAS, trait, and behavioral resilience, the total score

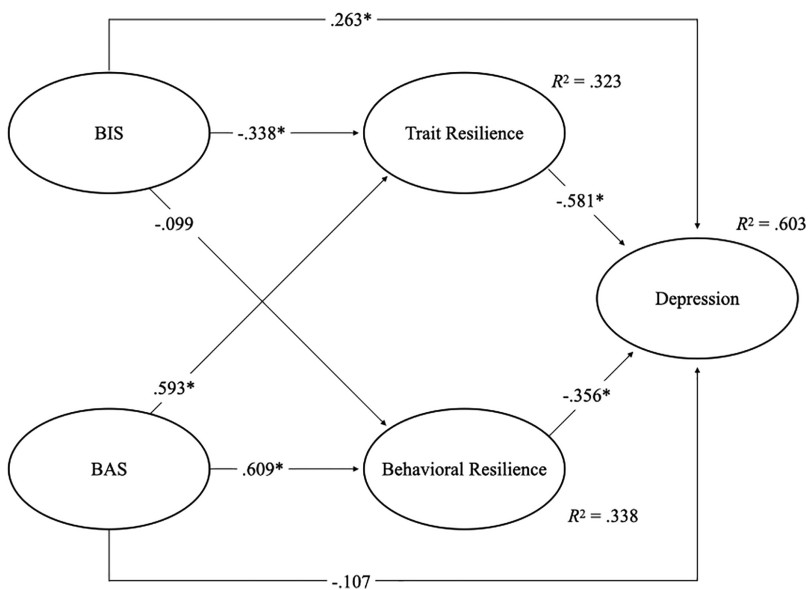

**Figure 2 Model estimation and its coefficients.** BIS, Behavioral Inhibition System; BAS, Behavioral Activation System.

of each subscale was used as an indicator. As depression and BIS were measured on a unidimensional scale, we applied a three-parcel-each-factor approach to preventing estimation bias (*Matsunaga, 2008*). The indices of fit of the model were good or sufficient ($\chi^2 = 562.911$, $df = 96$, $p < 0.001$; CFI = 0.925; NFI = 0.913; RMSEA = 0.073, 90% CI [0.067, 0.079]; SRMR = 0.066). All estimates of observed variables toward latent variables were significant ($p < 0.001$). Figure 2 presents the model of these variables.

According to the tested model, BIS influenced only trait resilience ($\beta = -0.338$, $p < 0.001$), whereas BAS influenced both trait and behavioral resilience ($\beta = 0.593$, $\beta = 0.609$, $p < 0.001$; respectively). The results showed that BAS positively predicted both trait and behavioral resilience, although behavioral inhibition negatively predicted only trait resilience. Next, trait resilience significantly affected depression ($\beta = -0.581$, $p < 0.001$). Similarly, behavioral resilience also significantly affected depression ($\beta = -0.356$, $p < 0.001$). These results demonstrated that both trait and behavioral resilience negatively predicted depression.

To investigate the indirect effects of BIS/BAS on depression *via* resilience, mediation analyses were conducted (Table 3). The mediation analyses were conducted in all pathways where BIS/BAS affected depression *via* trait and behavioral resilience, except the path where BIS involved depression mediating with behavioral resilience because of the non-significant effect of BIS toward behavioral resilience in the SEM. The analysis for BIS and depression mediated with trait resilience showed a significant indirect effect (estimate = 0.058, 95% CI [0.001, 0.118]), while the analysis for BAS and depression mediated with trait resilience also showed the significant indirect effect of trait resilience (estimate = −0.158, 95% CI [−0.191, −0.127]). Finally, the mediation effect of behavioral resilience in relation to BAS and depression was conducted. The result showed a significant mediation effect on behavioral resilience (estimate = −0.093, 95% CI

**Table 3 The results of mediation analyses.**

| Type | Effect | Estimate | *SE* | *t* | 95% CI | |
|---|---|---|---|---|---|---|
| | | | | | Upper | Lower |
| Direct | BIS -> Depression | 0.197 | 0.036 | 5.535* | 0.127 | 0.266 |
| Direct | BIS -> Trait resilience | −0.181 | 0.075 | 2.411 | −0.327 | −0.034 |
| Direct | Trait resilience -> Depression | −0.319 | 0.015 | 20.950* | −0.350 | −0.290 |
| Indirect | BIS -> Trait resilience -> Depression | 0.010 | 0.005 | – | 0.000 | 0.020 |
| Direct | BAS -> Depression | −0.350 | 0.021 | 1.657 | −0.076 | 0.006 |
| Direct | BAS -> Trait resilience | 0.501 | 0.037 | 13.493* | 0.428 | 0.573 |
| Direct | Trait resilience -> Depression | −0.315 | 0.017 | 18.720* | −0.348 | −0.282 |
| Indirect | BAS -> Trait resilience -> Depression | −0.028 | 0.003 | – | −0.033 | −0.023 |
| Direct | BAS -> Depression | −0.01 | 0.024 | 4.134* | −0.147 | −0.052 |
| Direct | BAS -> Behavioral resilience | 0.354 | 0.024 | 14.943* | 0.307 | 0.400 |
| Direct | Behavioral resilience -> Depression | −0.263 | 0.030 | 8.866* | −0.321 | −0.205 |
| Indirect | BAS -> Behavioral resilience -> Depression | −0.016 | 0.002 | – | −0.021 | −0.012 |

**Notes:**

BIS, Behavioral Inhibition System; BAS, Behavioral Activation System.

* $p < 0.001$.

[−0.119, −0.069]). These results suggested that the trait resilience mediated the relationships between BIS/BAS and depression, and behavioral resilience mediated the relation between BAS and depression.

## DISCUSSION

Our study aimed to test a hypothesized model on the effects of BIS/BAS on depressive symptoms *via* trait and process resilience. The fit indices of the tested model were good, and all estimates of the observed variables for each latent variable were significant. These suggested a good fit of our hypothesized model with the current dataset. Further, almost all of the variables yielded substantial estimates. Overall, we confirmed the relations among the following: BIS/BAS affected trait and behavioral resilience, and trait and behavioral resilience affected depressive symptoms. These results coincided with previous findings of the association of depression with BIS/BAS (*e.g.*, *Markarian et al., 2013*; *Mellick, Sharp & Alfano, 2014*; *McFarland et al., 2006*) and with resilience (for a review, see *Southwick, Vythilingam & Charney, 2005*). Our study cross-sectionally investigated resilience and BIS/BAS; we found the mediation effects of resilience in the relation between BIS/BAS and resilience. In addition, our findings revealed the relations between BIS/BAS and depressive symptoms in terms of the trait and process aspects.

Consistent with previous studies (*Mellick, Sharp & Alfano, 2014*; *Sportel et al., 2011*), our results demonstrated the relation of both BIS and BAS to depression. The results of our model estimation showed that both BIS and BAS directly influenced depressive symptoms: BIS increased depressive symptoms, whereas BAS decreased them. Based on the theoretical framework of Response Sensitivity Theory (*Gray & McNaughton, 2000*), BIS can lead to avoidance and escape-related behavior with negative affect. In contrast, BAS can lead to motivation and goal-directed behavior with positive affect. Thus, our

results indicate that high BIS and low BAS would be related to increased depression. Indeed, *Wu et al. (2021)* reported that high BIS predicts depressive symptoms in adolescents. *McFarland et al. (2006)* showed that BAS deficits predict worse depression in major depressive disorder. In comparing the standardized effects of BIS and BAS on depressive symptoms as absolute values, our analysis showed that BIS ($\beta = 0.279$) more strongly predicted depressive symptoms than BAS ($\beta = -0.179$). However, several studies investigating BIS and BAS cross-sectionally have reported the reverse: BAS more strongly indicates depressive symptoms in major depressive disorder (*Kasch et al., 2002*) and substance abuse disorder (*Xie et al., 2021*). Considering the concurrence of our results and those of a previous study (*Wu et al., 2021*) that targeted adolescents, the more significant influence of BIS than BAS on depression could be an adolescent-specific characteristic of depression symptomatology. Further study would be needed to elucidate how the effect of BIS/BAS on depression could differ with age.

Apart from the direct effect of BIS/BAS on depression, we also found the mediation effects of resilience. The results revealed the indirect effect of BIS *via* trait resilience toward depression (estimate = 0.058). Behavioral inhibition decreased individuals' trait aspect of resilience, subsequently decreasing depressive symptoms. The result suggested the buffering effect of trait resilience, highlighted in depression literature (*e.g.*, *Bitsika, Sharpley & Bell, 2013*; *Haeffel & Vargas, 2011*; *Waugh & Koster, 2015*). Thus, trait resilience can decrease the pathway of behavioral inhibition temperature, represented as avoidant or withdrawal behaviors toward depression.

The results for the BAS pathway similarly showed the mediation effect of trait resilience. The mediation analysis revealed significant indirect effect of trait resilience on the relation to the BAS and depression (estimate: −0.158). This indicated that high BAS is associated with a reduction in depressive symptoms and that the reduction is further enhanced by trait resilience. Alternatively, BAS could also increase trait resilience and consequently decrease depressive symptoms. Similar to BIS, the relation between BAS and depression has been reported (*Markarian et al., 2013*; *Mellick, Sharp & Alfano, 2014*). BAS has been correlated with trait resilience (*Genet & Siemer, 2011*; *Nam et al., 2018*). Our results were consistent with these studies, indicating that the behavioral activation temperature involved with positive emotion and reward responsiveness in mediating trait resilience is associated with depressive symptoms.

The results also revealed a negative effect of the behavioral aspect of resilience on depressive symptoms, suggesting that more resilience-related behavior could be linked to less depressive symptoms. Furthermore, the mediation analysis showed that behavioral resilience had significant indirect effect in relation to BAS and depression. As well as trait resilience, BAS could increase behavioral aspect of resilience and then buffer depression. However, the pathway of BIS to behavioral resilience obtained from model estimation was not significant ($\beta = -0.099$). Inconsistent with a previous study that BIS was negatively correlated to adaptive behavior such as self-reflection which also included in the concept of the behavioral resilience in this study (*Khosravani et al., 2020*), our results suggested that BIS did not predict behavioral resilience. Considering the high correlation between trait and behavioral resilience in this study ($r = 0.661$), as we mentioned later, it could be

possible that the division of resilience into behavioral and trait aspects may have partialed out the influence of behavioral resilience.

Our study, which distinguished the concept of resilience into trait and behavioral aspects, might provide insight into and add to the discussion on a trait–process resilience distinction (*Herrman et al., 2011*). Our study regarded trait resilience as a trait and behavioral resilience as a process. The model tested in this cross-sectional study showed that both trait and behavioral resilience significantly predicted depressive symptoms (trait: β = −0.581, behavioral: β = −0.356). The results suggested that the trait conceptualization of resilience can more buffer depressive symptoms than the process resilience. This suggestion was consistent with the research that has shown the buffering effect of either trait (*Anyan & Hjemdal, 2016*) and process (*Brailovskaia et al., 2018*) resilience against depression. However, our study's correlation between the trait and behavioral resilience was high (*r* = 0.661). Although there would be the limitation due to large overlap between trait and process resilience, we could conclude that trait resilience rather than behavioral resilience might have a buffering effect on depression, indicating the importance of trait but not process resilience.

Our study had several limitations that must be considered when interpreting our findings. First, given the cross-sectional nature of our study, the causal relations shown are mere estimations. The strict consequences of BIS/BAS, resilience, and depression merit further research using a longitudinal design. Second, there is no clear definition of the trait and behavioral aspects of the resilience division. In recent research, as a trait of resilience, ego-resiliency, which is involved with the control process of ego against stress (*Block & Block, 1980*; *Block, 2002*), has attracted attention in positive psychology. Empirical research has also shown the buffering effect of ego-resiliency on depression (*Mujeeb & Zubair, 2012*). The conceptualization of resilience, which remains ambiguous, needs to be further elaborated and divided into traits and behaviors. Lastly, our study was carried out during the COVID-19 pandemic. The model estimation could have been affected by the pandemic situation, such as school closures, fear of COVID-19 infection, and restrictions in the daily life of adolescents. Indeed, some studies have confirmed emotional response to COVID-19 infection with psychological distress (*Sugawara, Masuyama & Kubo, 2021*). Meanwhile, several studies have revealed the buffering effect of positive psychological factors on depression and anxiety (*Kubo, Sugawara & Masuyama, 2021*; *Masuyama et al., 2021b*). Including the effect of the long-lasting pandemic and its related psychological effect, further investigation of adolescents' mental health is needed.

## CONCLUSIONS

In this study, the BIS/BAS as a vulnerability and resilience as a protective factor toward depression in adolescents, which revealed their effects separately, were investigated cross-sectionally. In addition, this study divided resilience, which broadly examined and then resulted in ambiguous concepts, was divided into trait and behavioral resilience.
The model estimation showed the strong mediation effect of trait resilience, suggesting that the trait resilience could attenuate the effect of BIS on depression and that the trait resilience could enhance the preventive effect of BAS. The results in this study would

contribute to further intervention that targeted resilience and aimed to decrease depression because of the significance of trait resilience rather than process resilience.

### Funding
This work was supported by the Fukushima Prefecture Academic Foundation (No. 2021-15). The funders had no role in study design, data collection and analysis, decision to publish, or preparation of the manuscript.

### Grant Disclosures
The following grant information was disclosed by the authors:
Fukushima Prefecture Academic Foundation: 2021-15.

### Competing Interests
The authors declare that they have no competing interests.

### Author Contributions
- Akihiro Masuyama conceived and designed the experiments, performed the experiments, analyzed the data, prepared figures and/or tables, authored or reviewed drafts of the article, and approved the final draft.
- Takahiro Kubo conceived and designed the experiments, performed the experiments, analyzed the data, prepared figures and/or tables, authored or reviewed drafts of the article, and approved the final draft.
- Hiroki Shinkawa conceived and designed the experiments, analyzed the data, authored or reviewed drafts of the article, and approved the final draft.
- Daichi Sugawara conceived and designed the experiments, analyzed the data, prepared figures and/or tables, and approved the final draft.

### Human Ethics
The following information was supplied relating to ethical approvals (*i.e.*, approving body and any reference numbers):
Iryo Sosei University granted Ethical approval to carry out the study within its facilities (Receipt Number: 2021-09).

### Data Availability
The raw data is available in the Supplemental File.

### Supplemental Information
Supplemental information for this article can be found online at http://dx.doi.org/10.7717/peerj.13687#supplemental-information.

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
