# Peer review of "The roles of trait and process resilience in relation of BIS/BAS and depressive symptoms among adolescents"

_PeerJ, doi:10.7717/peerj.13687_

## Round 0.1 · original submission · Minor Revisions

Dear authors,

The manuscript needs some revisions to improve. Respond point by point to the reviewers' comments and resubmit the manuscript.

Thanks!

Reviewer 1 ·

Basic reporting

no comment

Experimental design

no comment

Validity of the findings

no comment

Additional comments

See attached PDF

Annotated reviews are not available for download in order to protect the identity of reviewers who chose to remain anonymous.

Reviewer 2 ·

Basic reporting

no comment

Experimental design

Participants: It could be better to maximize the participants and add descriptions of relevant demographic characteristics in detail, such as gender.
Procedures: Suggest specify the date range of data/survey collection, considering the depressive symptoms would be different at different test times (month/day/year). And the study mentioned it was conducted during the COVID-19 pandemic, so it’s better to describe the pandemic where participants were.
Results: provide the results with more details, including the description data of participants in the tables or figures.
For instance: Table 1. should add the note of *p< .05, **p< .01, ***p< .001.

Validity of the findings

no comment

Additional comments

no comment

Annotated reviews are not available for download in order to protect the identity of reviewers who chose to remain anonymous.

·

Basic reporting

1.The paper requires extensive language editing. Please ensure in revising your manuscript that this is attended to.
2. It is required to provide more evidence in the introduction that supports the hypotheses and to expand the findings on the role of resilience in the model.
3.Line 207: Please carefully check the symbol in this line. Currently 'ps<.001' is wrong.

Experimental design

no comment

Validity of the findings

1.It is recommended to include the x2/df and SRMR values, in the results of the structural equations.
2.It is recommended to give more details about demographic characteristics of the study sample, for example the gender and the percentage of reported symptoms of depression.
3.Figure 2 Please carefully check all the path coefficients are standardized values in this figure. Currently 'the path coefficient from trait resilience to depression ' is greater than 1 which means it is unstandardized.
4.To test the mediation effect, the bias-corrected 95% confidence interval (CI) should be calculated with 1,000 bootstrapping samples. If the 95% CI of the indirect effect does not include 0, a significant mediation effect can be established. Please use this method to test the mediation effect in this study.

·

Basic reporting

English is clear but should be revised. Punctuation sould also be revised
Literature references are updated and appropiately quoted. The background of the study is sufficiently convincing
The authors may wish to quote V. De Pascalis regarding the possible relation between BIS/BAS and optimism, which could be studied in further ressearch by the same group

The article structure, figures and the table are OK. Raw data availability is obvious as authors analyzed an open dataset

Experimental design

No comment

Validity of the findings

No comment

Additional comments

Please check English and punctuation across the text

---

## Round 0.2 · accepted · Accept

Dear authors,

Thanks for the replies to the reviewers' comments.
The manuscript is ready to be published on PeerJ.